# An Anatomy-Guided, Stepwise Microsurgical Reconstruction of a Posteriorly Projecting ICA–PCoA Aneurysm Beneath the Optic Apparatus: A Detailed Operative Sequence

**DOI:** 10.3390/diagnostics16010124

**Published:** 2026-01-01

**Authors:** Matei Șerban, Corneliu Toader, Răzvan-Adrian Covache-Busuioc

**Affiliations:** 1Department of Neurosurgery, “Carol Davila” University of Medicine and Pharmacy, 050474 Bucharest, Romania; 2Puls Med Association, 051885 Bucharest, Romania; 3Department of Vascular Neurosurgery, National Institute of Neurology and Neurovascular Diseases, 077160 Bucharest, Romania

**Keywords:** ICA–PCoA aneurysm, posterior communicating artery, microsurgical clipping, carotid–optic corridor, two-clip reconstruction, perforator-sparing, anterior choroidal artery, aneurysmal subarachnoid hemorrhage, delayed cerebral ischemia, digital subtraction angiography

## Abstract

**Background**: Posteriorly directed aneurysms at the internal carotid–posterior communicating artery (ICA–PCoA) junction concentrate technical risk at the posteromedial neck where the PCoA origin and perforators exist beneath the optic apparatus. Our aim was to describe, in a reproducible fashion, an anatomy-driven sequence in the management of a ruptured ICA–PCoA aneurysm that visualized the posterior wall and a closing line parallel to the PCoA axis and which is placed within contemporary practice. **Case Presentation**: This is a single case study employing predetermined surgical techniques demonstrating a reproducible method of anatomical microsurgery applied to a posterior projecting ICA-PCoA aneurysm. The authors describe a 62-year-old female who was stabilized by nimodipine and aggressive blood pressure control in the systolic range 140–160 mmHg after an aneurysmal subarachnoid hemorrhage. Diagnostic contrast catheter angiography showed a left ICA-PCoA aneurysm of 13.1 × 10.0 mm at the base with a neck of 4.3 mm projecting posteriorly into the carotid–optic cistern. Complete adherence to a protocol of staged techniques was employed for the operation, as detailed below. Step 1: Early cisternal decompression requiring total and immediate relaxation of the temporal lobe, rapidly opening up the carotid–optic anatomical window. Step 2: Circumferential dissection about the neck of the aneurysm permitting definition of the true posteromedial wall and definition of the perforator territories and anterior choroidal territories. Step 3: Brief but effective ICA proximal quiescence (58 s) permitting clipping under direct vision. Step 4: Staged closure of two clips with the closing line of the clips orientated parallel to the axis of the PCoA with maintenance of the diameter of all parent vessels, the origin of the PCoA and the integrity of the perforators. Urgent postoperative digital subtraction angiography (DSA) study showed complete exclusion of the aneurysm with no alteration in flow characteristics, and 3 months later DSA studies again showed permanent obliteration and patency of those branches. The immediate DSA demonstrated complete exclusion of the aneurysm with patent supraclinoid ICA caliber and PCoA ostium, the anterior choroidal artery was preserved; no angiographic vasospasm was identified. The postoperative course was uncomplicated; there was no hydrocephalus, seizure disorder or delayed ischemia. At discharge and three months postprocedure the patient was neurologically intact (Modified Rankin Scale 0). Non-contrast cranial CT (three months) demonstrated stable clip position and no hemorrhagic or ischemic sequelae. **Conclusions**: In posteriorly projecting ICA–PCoA aneurysms that are disturbed beneath the optic apparatus, an anatomy-guided strategy—early cisternal decompression, true posteromedial neck exposure, brief purposeful quieting of the proximal ICA and two-clip closure parallel to the PCoA in selected cases—may provide the opportunity for durable occlusion whilst the physiology of branching is preserved. We intend for this transparent description to be adopted, refined or discarded based on local anatomy and practice.

## 1. Introduction

Intracranial aneurysms of the internal carotid artery (ICA)–posterior communicating artery (PCoA) junction represent one of the most common locations of aneurysmal subarachnoid hemorrhage (SAH), accounting for as much as one quarter of ruptured lesions in large series [1,2,3]. They occur at the junction of the supraclinoid ICA and the origin of the PCoA, a zone of hemodynamic stress and curvature of the vessel in a narrow neurovasculature corridor defined by the optic apparatus, the carotid artery, and the posterior wall, which is replete with perforators. The resulting aneurysms are very difficult, however, because even small differences in vector or corridor selection can compromise the anterior choroidal artery (AChA) and PCoA perforator, making them complex [4]. ICA–PCoA aneurysms that are directed posteriorly are constrained by the geometry of the carotid–optic window. Thus, the posterior wall has a tendency to slant in the direction of the posteromedial quadrant, existing in a field traversed by hypothalamic and thalamoperforating vessels. The surgical question is not one of size but of spatial relations—how can the posterior wall, perforators, and branching blood vessels be visualized and protected while maintaining a nerve-safe operative corridor? The surgical corridor is thus “safe,” a phrase extensively discussed by classic surgical atlases, the key determinant of outcome being the principle that visualization and gentle protection must precede clip commitment and that a sufficient line of sight to the posteromedial neck should be achieved, not forced [5,6].

Recent investigations have added to the anatomical awareness. It has been firmly established by early studies into aneurysm morphology and flow dynamics that aneurysm projections and the curvature of the parent vessel control the propensity to rupture rather more than the maximum diameter [7,8,9]. Most recently, radiomic and flow-dynamic studies have demonstrated that fetal-type posterior cerebral artery (fPCA) configurations positively alter the outflow geometry of the vessels and contribute to irregularly shaped dome morphology, explaining why ruptures of such aneurysms are more common and providing a better explanation for differences in flow diversion reaction during turbulent flow characteristics in some cases [2]. Clinical data from the microsurgical and endovascular series also conclusively confirm that persistent obliteration is dependent upon adaptation of the corridor to the specific individual vascular anatomy, and it is therefore unnecessary to adhere to a dogma [10].

In line with these developments, the strategic operative approach also evolves. This “corridor-guided” philosophy—to render apparent anatomical guilt regardless of tradition—has been habitually correlated with improved functional outcomes, the integrity of branches and general conservatism of the approach. Based on such principles, early cisternal decompression and retraction-free dynamic dissection ensure a low-pressure arena, expanded in the optic–carotid window, allowing expansion without traction injury [11].

Notwithstanding these important advances, the illustration of posteriorly directed ICA–PCoA aneurysms beneath the optic apparatus by detailed, stepwise dissection is still little discussed. The following report illustrates such a case with complete anatomical dissection. A 62-year-old woman with a ruptured ICA–PCoA aneurysm underwent early cisternal decompression, followed by circumferential definition of the neck, transient quieting of the proximal ICA and two-clip reconstruction in a stepped fashion, with the closing line parallel to the PCoA axis. Because this description is based on a single patient, no attempt is made to generalize, but it is merely to illustrate an anatomical-driven logic of reproducible incision and closure aptly demonstrative of such cases. Yes, of course, all is well in life, whatever happens under the skin.

## 2. Case Presentation

### 2.1. Clinical Presentation and Emergency Stabilization

A 62-year-old woman, presenting as a result of a hyperacute, maximal thunderclap headache—holocranial with left fronto-orbital predominance, which occurred approximately one hour prior to her presentation—had previously experienced hypertensive retinopathy and silent ischemic heart disease and had stage II arterial hypertension. Immediately following the onset of the headache she began to experience photophobia, phonophobia, intractable emesis and an intense autonomic surge (systolic blood pressure 198 mmHg, diastolic blood pressure 108 mmHg, pulse rate 104 beats per minute, respiratory rate 22 breaths per minute, oxygen saturation 96% in room air, body temperature 36.9 °C) and a transient gastroesophageal activation causing systolic spikes >210 mmHg, resulting in a suspected hemorrhagic physiology that needed to be blunted without compromising her ability to assess her neurological status.

We immediately implemented hemodynamic control measures: a 30° elevation of the bed, midline cervical alignment to facilitate venous egress, coached normocapnia (to avoid hyperventilating) and a continuous nicardipine infusion (3 mg/h, with adjustments in 0.5–1.0 mg/h increments) to maintain a systolic blood pressure (SBP) of 140–160 mmHg. We continuously monitored SBP via arterial waveform tracing. The time-stamped interventions included initiation of the nicardipine infusion at 7 min, administration of fentanyl 50 mcg IV at 9 min to decrease the frequency of the emetic-induced surges in blood pressure without decreasing the cortical sensory feedback and administration of ondansetron 4 mg IV at 12 min, while monitoring QTc intervals in real time.

At 12 min postarrival, the patient remained conscious but pathological (Glasgow Coma Scale = 14: E4 V4 M6). She displayed obvious evidence of meningeal irritation—rigidity at 20° passive flexion and positive Kernig (~135° knee extension) and Brudzinski signs, and jolt accentuation produced a reproduction of the thunderclap headache with vegetative flare. Her headache intensity was NRS 10/10, immediate photophobia was present and light tactile sweep along the left V1 dermatome reproduced reproducible periorbital allodynia and retro-orbital pressure during prolonged upgaze. From the neurological examination the patient’s state corresponded to a Hunt–Hess grade II and a World Federation of Neurosurgical Societies (WFNS) grade I, indicating a stable condition with an increased risk of deterioration in the first few hours after ictus.

### 2.2. Neurological Examination

Due to the severe photophobia of the patient, cranial nerve testing was performed using controlled stimulation to minimize the potential elevation of intracranial pressure. Gradual acclimatization allowed for pupillary probing without causing blood pressure surges due to stress. Tracking resulted in brief, non-localized saccadic intrusions, while deep palpation of the left trochlear fossa resulted in intense retro-orbital pain, again supporting irritation of the ipsilateral carotid cistern. Pain-limited motor testing demonstrated bilateral hypometria on finger–nose and heel–knee–shin testing without lateralized dysmetria. Due to concerns of inducing hypertensive spikes, formal drift testing was delayed.

Each successive step in the evaluation was weighed against sympathetic output to maximize the quality of the diagnostic information obtained. Overall, the evaluation demonstrated a meningeal-dominant, pain-restricted pattern of symptoms rather than a toxic/metabolic stupor and validated the clinical diagnosis of aneurysmal subarachnoid hemorrhage (SAH).

### 2.3. Laboratory and Cardioneural Correlation

Initial laboratory values demonstrated a stress leukocytosis (WBC 13.2 × 10^9^/L) with a mild CRP elevation (4 mg/L), typical of neurogenic inflammation. The serum sodium concentration was 133 mmol/L, indicative of early subneuroendocrine hyponatremia secondary to SAH. Glucose levels were elevated (168 mg/dL), consistent with stress hyperglycemia. QTc interval lengthened to 486 ms, and there were anterior T-wave inversions (V3–V5). High-sensitivity troponin was slightly elevated (42 ng/L), and BNP was also slightly elevated (188 pg/mL). Point-of-care echocardiogram showed mild apical hypokinesis, however LVEF was well-preserved (>50%), demonstrating neurogenic myocardial stunning. This information was utilized to inform a pressor-sparing, balanced fluid and analgesic management strategy.

### 2.4. Risk Mitigation in the Acute Phase and ICU Protocol

In the first hours, our management strategy focused on mitigating three primary risks—rebleeding, acute hydrocephalus and malignant intracranial hypertension (ICP). Nicardipine was carefully titrated to provide adequate rebleed prevention, while minimizing the need for significant analgesics to reduce sympathetic surges. Hydrocephalus surveillance included frequent high-frequency neurochecks (every 15 min × 2, and then every hour) and pupillometry escalation thresholds (intervention for Neurological Pupil Index ≤ 1 mm). An external ventricular drainage (EVD) plan was developed, including access via the right frontal Kocher point (1 cm anterior to the coronal suture, 2.5–3 cm lateral to midline), a catheter trajectory perpendicular to the foramen of Monro, catheter length 5.5–6.5 cm and initial drainage level +10 cm H2O. Protocol for malignant ICP: CSF diversion when indicated, followed by hyperosmolar therapy (goal euvolemic and sodium 140–145 mmol/L) while avoiding sedatives that may obscure neurological progression. Nimodipine 60 mg q4h (30 mg q2h if pressure-sensitive) and continuous nicardipine maintained CBF. Fentanyl and ondansetron were used for analgesia and antiemesis, respectively. One gram of tranexamic acid was administered IV and discontinued once the aneurysm was excluded. Levetiracetam (1 g IV bolus, then 500 mg BID) was used for seizure prophylaxis. Both chemical and mechanical DVT prophylaxis were initiated immediately after aneurysm exclusion.

The airway was intentionally managed without intubation to maintain a continuous neurological signal. No sedatives or long-acting antiemetics were used. Intake-output monitoring was strictly implemented to detect early natriuretic diuresis and blood products were cross-matched in preparation for anticipated surgical intervention.

### 2.5. Imaging Findings and Morphometric Analysis

Catheter digital subtraction angiography (DSA) in anteroposterior oblique and lateral views (Figure 1A,B), supplemented by 3D rotational angiography (Figure 2A–D), demonstrated a left ICA-PCoA junction aneurysm originating from the communicating (C7) ICA segment and extending posteriorly into the carotid–optic cistern. Three-dimensional quantitative measurements of the aneurysm size demonstrated a maximum sac height of 13.1 mm, maximum sac width of 10.0 mm and maximum neck diameter of 4.3 mm, producing an aspect ratio of 3.06 and dome-to-neck ratio of 2.34. The aneurysm had a bilobed morphology with a smaller posterior superior daughter sac located along the posteromedial neck quadrant precisely adjacent to the PCoA origin and its associated perforator field—where even minimal posterior clip torque would pose a threat to the thalamoperforators.

Combining the geometric and spatial parameters of this aneurysm with the relationship of the optic and carotid arteries defined a posteriorly projecting aneurysm below the optic apparatus, making it suitable for a traditional pterional craniotomy with early cisternal decompression, 360-degree delineation of the neck and a clip line parallel to the take-off of the PCoA.

### 2.6. Preoperative Angiographic Findings

Angiography revealed a 13.1 × 10.0 mm bilobed posteriorly projecting aneurysm of the left ICA arising from the C7 (communicating) segment with a 4.3 mm neck that extended into the carotid–optic cistern with a small portion of the aneurysm extending into the posteromedial perforator bed. The PCoA was identified as a well-formed artery originating directly from the neck of the aneurysm without any branches or involvement of the surrounding arteries. The proximal and distal portions of the ICA were found to be smooth and of equal diameter, without stenosis or other abnormal features.

Reconstructions allowed visualization of the anatomic relationship of the aneurysm to the surrounding structures including the optic nerve, the anterior choroidal artery, and the oculomotor nerve. These reconstructions demonstrated that there would be a small margin of safety above the aneurysm due to its location immediately below the optic nerves and the anterior portion of the optic chiasm, thereby limiting the degree of upward displacement of the aneurysm dome before it came into contact with the optic nerves.

Additionally, the bilobed nature of the aneurysm resulted in increased risk to the surrounding perforators, which were located in the posteromedial portion of the aneurysm. Therefore, the closing tangent needed to be oriented in a manner that would allow for complete removal of the aneurysm dome without encroaching on the PCoA bed or any of the surrounding perforators. Given the size of the neck (4.3 mm), a staged closure technique using two clips was planned to minimize the amount of force used in the closure of the aneurysm neck, thereby minimizing the potential for damage to the parent vessel and its branches. Quietening of the ICA temporarily was planned to occur only during the final stages of closure of the aneurysm neck to assist in reducing the turgor of the blood within the ICA and to aid in the positioning of the second clip.

### 2.7. Operative Positioning and Surgical Exposure

The patient was placed supine on the operating table with a three-pin Mayfield head holder and the malar eminences at the top of the operative field. The patient’s head was positioned so that the left carotid–optic axis was in a straight, shallow path to allow for gravity-assisted countertraction of the frontal lobe.

A curved frontotemporal skin incision was made, and the frontal branch of the superficial temporal artery was preserved by dissecting between the layers of the scalp. The temporalis muscle was reflected inferiorly on a subfascial cuff to provide adequate exposure of the temporal fossa. A standard pterional craniotomy was performed to expose the supraclinoid internal carotid artery. The sphenoid ridge was drilled down to the level of the meningo-orbital band to reduce the working distance to the supraclinoid carotid artery and to flatten the angle of approach into the carotid–optic window. The dura mater was opened in a curvilinear fashion toward the sphenoid wing and tacked to the underlying bony edges to create a vascularized, self-retaining dural flap. No fixed retractors were used during the case.

After opening the carotid cistern, cerebrospinal fluid (CSF) began draining freely and the frontal lobe relaxed significantly. Subsequent fenestration of the diencephalic and mesencephalic leaves of the membrane of Liliequist created a continuous drainage pathway into the interpeduncular cistern, creating a gravity-dependent, low-pressure operative field without the need for retractors.

Two separate operative corridors were developed to facilitate exposure of the aneurysm:A subfrontal corridor was developed to visualize the supraclinoid ICA and to identify the optic nerve. Arachnoid adhesions were dissected to widen the carotid–optic window to approximately 6–8 mm.A limited sylvian fissure splitting procedure was developed to provide additional vertical exposure. This procedure involved sequential dissection of the sylvian fissure to preserve superficial sylvian veins.

Using these two corridors, both the anterior choroidal artery (AChA) and the oculomotor nerve could be visualized. The AChA was identified as a thin branch arising distal to the communication segment of the ICA. The oculomotor nerve was identified as it passed through the carotid–oculomotor triangle, providing a clear definition of the inferior and posterior boundaries of the operative corridor.

### 2.8. Microsurgical Dissection and Aneurysm Reconstruction

Dissection of the communicating ICA was begun at the point where the PCoA took off, and the PCoA ostium was brought into view. The neck of the aneurysm (~4.3 mm) arose from the shoulder of the aneurysm dome, supporting a bilobed dome (~13 mm height, 10 mm width) directed posteriorly. The posteromedial quadrant represented the zone of greatest risk due to the convergence of the thalamoperforators and the PCoA bed. All adhesions were dissected sharply under irrigation, taking care to avoid traction on either the perforators or the sac of the aneurysm. Clot was teased free from the laminated surface of the aneurysm dome with the back edge of a bayonet dissector. Cottonoids were used to protect both the optic nerve and the PCoA origin when making angle adjustments. The neck of the aneurysm was successively dissected circumferentially, one quadrant at a time, until the posterior wall of the aneurysm was clearly visible. No looping maneuvers were used to develop the posterior wall of the aneurysm, but instead, exposure was developed by continuous visualization of the posteromedial bed.

Proximal ICA control was reserved for reconstruction of the aneurysm. Once all of the surrounding structures had been separated from the plane of closure, temporary clipping of the proximal ICA was applied for 58 s, during which time systemic blood pressure was maintained in the range of 140–160 mm Hg. During the “quiet” period, a curved 9 mm Yasargil clip was positioned parallel to the PCoA axis, with the feet of the clip seated off of the posteromedial perforators and the hinge of the clip lateral to the optic apparatus. A small residual (“dog ear”) was obliterated with a 5 mm finishing clip, completely sealing the aneurysm without causing any alteration of the PCoA or AChA courses or any torsion of the posterior wall. After flow restoration, the entire construct was examined microscopically to verify that the ICA remained smooth and of normal diameter; the PCOA ostium pulsed and remained blanchless; the AChA remained pink and patent; and all thalamoperforators remained intact and exhibited pulsation. Microscopic yaw was also used to verify that the posterior wall of the aneurysm was free and that the clip did not chatter.

Hemostasis was accomplished using a fine-point bipolar coagulator on the arachnoid leaves. Energy was not applied near the perforators.

### 2.9. Closure and Postoperative Verification

The relationships between the clips and their neighboring structures were measured precisely: clip-to-PCoA ostium clearance was 2–3 mm; the closing line of the clips was parallel to the PCoA axis; the hinges of the clips were lateral to the optic nerve; and the blades of the clips were below the optic apparatus within the carotid–optic window. Any further instrument movement was severely limited by the carotid–oculomotor triangle. Before each clip-angle adjustment, the origins of the AChA and PCoA were reidentified. The dura was closed in a watertight fashion, and the bone flap was replaced and secured with a short epidural vent repositioned away from the craniotomy edge to prevent epidural hypertension.

The patient was extubated on the day of her discharge from the ICU and was awake and alert (GCS 15) shortly thereafter. Her cranial nerve examination was essentially normal, with full ocular motility, no ptosis, no diplopia, no anisocoria and no allodynia over the distribution of the left V1. Motor function was normal, and coordination was non-lateralizing. Nimodipine 60 mg PO Q4hr was continued, and the patient was kept within a systemic blood pressure range of 140–160 mmHg. Analgesia and antiemetics were administered in small aliquots in order to keep examinations brief. No CSF diversion was necessary, and hydrocephalus surveillance was continued via hourly neurochecks and pupillometry for 48 h. The epidural vent drained very little and was removed uneventfully.

### 2.10. Outcome and Follow-Up

At 24 h postoperatively, follow-up DSA was performed to evaluate the technical success of the procedure. Complete exclusion of the aneurysm was verified, as was preservation of the ICA caliber and the patency of both the PCoA and AChA. Symmetrical hemispheric opacification was observed, and no evidence of delayed transit was noted (Figure 3).

Subjectively, the patient reported improvement in headaches from NRS 10/10 preoperatively to 3/10 by postoperative day 2. Photophobia and nausea resolved entirely. She experienced no delayed ischemic neurological deficit, no language deficits and no ocular dysfunction. She ambulated on day 2, transitioned to oral nimodipine after achieving a swallow clearance and maintained stable telemetry as her catecholamine fluctuations normalized. Her wound and bone flap remained intact without evidence of CSF leakage. She was discharged home neurologically intact (MRS 0, GOS 5) on postoperative day 5.

At three months, she returned headache-free and fully functional, with no visual or visual–motor sensations, complaints related to gaze or other sensory complaints. Tandem gait was steady; coordination was fluid; there were no exertional triggers or sentinel pains. At that visit, the non-contrast head CT showed a stable clip complex, centered in the left suprasellar/carotid–optic cistern; patent basal cisterns; age-appropriately sized ventricles without ventriculomegaly or transependymal seepage; and without hemorrhagic or ischemic character alterations (Figure 4). These findings aligned with her normal, walks out of health care and an appropriate health-related quality of life (HRQoL).

This aneurysm occurred posterior to the PCoA take-off, under the optic apparatus and adjacent to the posteromedial perforator field. We needed, in order: cisternal decompression before retraction, 360° line-of-sight neck definition, a closing line that was kept parallel to the PCoA and a brief proximal quiet only for the vital seconds of reconstruction. Findings on postoperative and follow-up images offered the same: an angiographic cure, preserved branch and perforator patency and complete recovery of clinical function which we have maintained at three months. We present this case with the modest aim of illustrating a practical, anatomy-guided sequence that may be reproducible in ICA–PCoA aneurysms of similar configuration.

## 3. Discussion

We approached this ruptured ICA–PCoA aneurysm as an anatomic and physiologic problem whose solution depended on respecting the local branching geometry, the microvascular neighborhood of the PCoA origin and the clinical realities of an unsecured dome in the acute phase. Our preoperative analysis emphasized that the PCoA take-off lay tangential to the posteromedial quadrant of the neck and that perforators in this corridor would be the rate-limiting structure; this directly informed the plan for early cisternal decompression, exposure of the carotid–optic window and staged clipping that closed parallel to the PCoA axis rather than across it. That logic is consistent with contemporary operative series that treat PCoA lesions as “branch-anchored” rather than pure sidewall disease and report complete occlusion with deliberate posteromedial visualization rather than force-vector clipping on a partially seen neck. In a 2024 single-center clinical series using a subtemporal strategy, complete clip occlusion was achieved in all cases with acceptable morbidity; although our corridor differed, the unifying principle—gain depth and line of sight to the posteromedial neck before committing the blades—remains the same [5].

In this location, temporary proximal control may be lifesaving, yet it is never biologically neutral. We tried to minimize ischemic time and clamp force and to reserve flow arrest for the instant of neck closure. The caution is warranted: a 2025 aneurysm-clipping series using intraoperative diagnostic cerebral angiography (ioDCA) showed that “microsurgically quiet” temporary clips can still produce parent-artery vasospasm or endothelial injury that only angiography reveals, a finding that rationalizes preplanned, short-duration occlusion and meticulous clip choreography. Work published in *Acta Neurochirurgica* in 2024 added a second, practical insight—contrast stasis within a tiny residual pouch on ioDSA predicted spontaneous thrombosis and durable occlusion—suggesting that not every pinpoint remnant demands further manipulation if the hemodynamics are favorable [12]. Together, these data support a conservative, eyes-open philosophy: use temporary clips sparingly, verify the parent vessel and let benign remnants with delayed washout alone [12,13]. In order to place these cautions into a wider, meaningful context, Table 1 seeks to summarize and interpret recent original studies that inform the everyday dilemma of ICA–PCoA aneurysms (including decisions about preoperative geometry and corridor selection, temporary occlusion and verification imaging and risk of early delayed cerebral ischemia (DCI) and system factors). We are thinking of it as a simple, decision-grade roadmap rather than a complete article review.

Our postoperative verification choices aligned with what is emerging about imaging performance after clipping. Three-dimensional digital subtraction angiography (DSA) remains the most sensitive test for small neck remnants and branch compromise immediately after surgery, and multimodality comparisons in 2024 confirmed 3D-DSA as the current gold standard for detecting tiny residuals. That said, photon-counting CTA has matured rapidly; an open-access 2025 series that included clipped and coiled aneurysms showed promising artifact handling and residual detection, while subspecialty reviews in 2025 argue that high-resolution CTA may be the preferred non-invasive modality for long-term surveillance when one or two modern titanium clips are present. We followed this arc—confirming obliteration and branch patency with catheter angiography early, then using CT-based surveillance to document stability thereafter [19,20,21].

Regarding neurological recovery biology after a PCoA rupture, the perioculomotor neighborhood matters. Oculomotor dysfunction is a hallmark of mass effect in this compartment, but modern datasets complicate the old dogma that clipping categorically outruns coiling for nerve recovery. A 2024 original study examining PCoA aneurysms with concomitant third-nerve palsy found that recovery depends on multiple factors—aneurysm size, acuteness of intervention and dome–nerve geometry—rather than on technique alone. We aimed to decompress the cistern early, keep retractors dynamic rather than fixed and avoid posterior wall torque during closure—judgment calls designed to reduce traction and pulsatility on the nerve even in the absence of a frank palsy [22].

Downstream cerebral ischemia after aSAH remains the determinant of late outcome. We structured our monitoring with the idea that DCI risk is not uniform and can be learned from the initial scan and clinical stream, even before vasospasm is angiographically obvious. Two 2024–2025 multicenter machine-learning studies moved this conversation forward: one integrated non-contrast CT radiomics with clinical data and deep-learning features and showed that an imaging-informed classifier can predict DCI more accurately than conventional scores; a second, larger 2025 study built an admission CT radiomics nomogram in 377 patients that stratified DCI risk early. While these tools are still translational, they underscore why consistent euvolemia, nimodipine exposure and frequent exam windows are not simply “protocol” but risk-matched care [15,16].

The adjunctive pharmacology during the unsecured interval remains somewhat enigmatic. The most recent large-scale randomized clinical trial of ultra-early, short course administration of tranexamic acid (TXA) resulted in fewer instances of rebleed; however, there were no differences in six-month functional status between treatment groups [23]. A post hoc analysis of the ULTRA dataset published in 2024 suggests that the protective effect of TXA is highly time-dependent, with greatest efficacy when administered immediately after onset of ictus; however, the lack of improvement in function indicates a need for judicious application and timely securing of the aneurysm [24]. We used TXA as a temporary bridging agent and not as a therapeutic end point. Beyond the vasospasm prevention afforded by nimodipine, the 2024 phase 2b clinical trial of long-acting nicardipine implants demonstrated a reduction in angiographic spasm and delayed cerebral ischemia (DCI). Early studies utilizing intrathecal nicardipine are continuing to assess its safety and feasibility as a method of local cerebrovascular drug delivery, which can increase bioavailability of the medication at the site of action while decreasing the risk of systemic hypotension. Although neither of these methods has been established as a standard of care, they demonstrate the utility of local cerebrovascular drug delivery as a means to improve the efficacy of medications utilized for vasospasm prophylaxis. IV milrinone is being studied as a rescue therapy for patients experiencing refractory vasospasm who fail to respond to blood-pressure-augmenting agents. Although preliminary and requiring prospective validation, the 2025 multicenter retrospective study demonstrating promising results with IV milrinone provides insight into the evolving management of patients with vasospasm. As such, our management philosophy includes maintaining a reliable background of nimodipine, preserving the neurologic examination and using investigative agents for refractory cerebrovascular physiology [18,24,25].

Epidemiology and predilection also matter because they contextualize what we see at the microscope. Contemporary case–control work across 1883 patients in 2024 found that women have higher odds of rupture after adjusting for location, size and vascular history, with an age interaction such that younger women remain at disproportionate risk; this observation dovetails with clinical experience that the PCoA complex, by virtue of its curvature and branch work, is over-represented among ruptured lesions. Our patient’s profile fits this signal [26].

Health-system structure influences outcome just as much as intraoperative nuance. A 2025 nation-level analysis from Brazil reported rising absolute costs of aSAH care over 2017–2022 driven by ICU days and procedure expenditures, while a 2025 German study demonstrated a clear volume–outcome relationship for ruptured aneurysm treatment—centers with higher annual caseloads achieved better survival and functional independence even after adjusting for case-mix. These data justify organized regionalization and consistent pathways that shorten the interval from ictus to aneurysm security and concentrate complex microsurgery in teams that do it often [26].

Finally, diagnostic front-end realities continued to evolve during our study period, which influences who reaches the operating room and when. The prospective SHED cohort across 88 UK emergency departments showed that a negative multislice CT within six hours of headache onset carries an exceptionally low posttest probability of SAH and that sensitivity remains high beyond six hours, albeit with a measurable decline; this validates fast imaging as a gatekeeper while reminding clinicians that time matters and that borderline presentations still need cerebrospinal fluid or vascular imaging depending on pretest risk. Such evidence reinforces why a door-to-decision culture is not merely efficient but clinically protective for patients with aneurysmal thunderclap [9].

In synthesis, we intended to show that the favorable arc in this case arose from ordinary decisions executed with unusual consistency: early hemodynamic containment that preserved the neurological signal, exposure that prioritized the posteromedial neck and PCoA origin, disciplined use of temporary control, verification with angiography when it most matters and a ward strategy tuned to the evolving science of DCI risk. Each step is portable to similar ICA–PCoA anatomies and, we hope, useful to others who face the same corridor and the same constraints.

## 4. Conclusions

This ruptured, posteriorly directed ICA–PCoA aneurysm was treated as an anatomy-based problem, where each decision was based on what could be seen, protected and confirmed at the posteromedially located neck. Our preoperative preparation for the surgery indicated that performing early cisternotomy would reduce pressure within the optic nerve sheath and enlarge the carotid–optic window; provide complete circumferential neck detail and an unobstructed posterior wall view to allow for safe closure; minimize ICA proximal occlusion time to maintain stability throughout the few seconds required to apply the clips; and provide a step-by-step (staged) reconstruction that would protect both the AChA and perforators. The final angiographic result and clinical outcome validated our microsurgical approach to safely treat this type of aneurysm in the posterior cerebral circulation with precision.

The primary determinant of the safety for posteriorly projecting ICA–PCoA aneurysms that are located below the optic nerves is direct visualization of the posteromedial neck and aligning the closure line with the axis of the PCoA, regardless of size or treatment method. When treating these types of aneurysms, the preoperative evaluation should include determining the size of the carotid–optic window, evaluating the presence of any outflow variants, and predicting how the various anatomical constraints will influence the pathway available for clipping and the geometry of the clips.

Future publications would greatly benefit by documenting the specific details of the aneurysm’s morphology, the variations present in the vessels involved, the relationship between the clips and the PCoA and AChA and the results of intraoperative perfusion studies. Collectively documenting these anatomy-based details will enable us to establish reproducible microsurgical criteria for the treatment of anatomically constrained ICA–PCoA aneurysms and ultimately define the situations in which a traditional pterional approach or alternative route provides the most protection.

From a practical perspective, this case illustrates that an anatomy-based, visualization-driven microsurgical approach to aneurysm treatment is a durable methodology that can remain effective as other approaches continue to evolve. The described sequence may serve as a model for other surgeons to utilize as a reproducible template to support their decisions and potentially improve the outcomes associated with the treatment of anatomically constrained ICA–PCoA aneurysms.

## Figures and Tables

**Figure 1 diagnostics-16-00124-f001:**
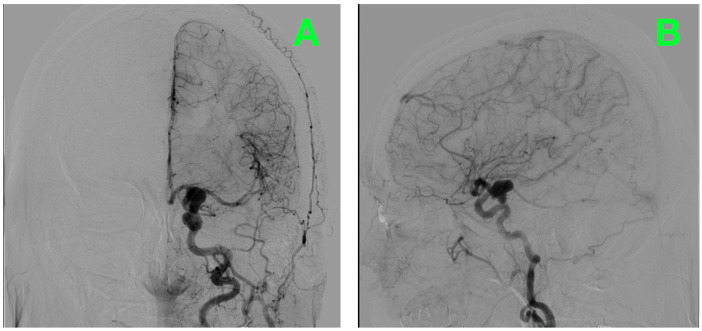
Diagnostic digital subtraction angiography of the left internal carotid artery. (**A**): Anteroposterior projection demonstrating a saccular aneurysm (arrow) arising from the ICA–posterior communicating artery (PCoA) junction on a caliber-preserved supraclinoid ICA. The PCoA is well-formed and originates at the aneurysm neck. On calibrated analysis the dome measures ~13 mm with a neck ~4.3 mm, projecting posteriorly into the opto-carotid cistern. Distal ACA and MCA branches opacify normally, with no angiographic vasospasm; the anterior choroidal artery opacifies separately and remains uninvolved. (**B**): Lateral projection confirming a posterior–superior dome trajectory with a bilobed contour. A compact perforator cluster is visualized posteromedial to the neck at the PCoA origin—an area corresponding to thalamoperforators and adjacent short circumflex branches—underscoring the need for a clip line that runs parallel to the PCoA take-off while preserving the anterior choroidal artery and perforator patency.

**Figure 2 diagnostics-16-00124-f002:**
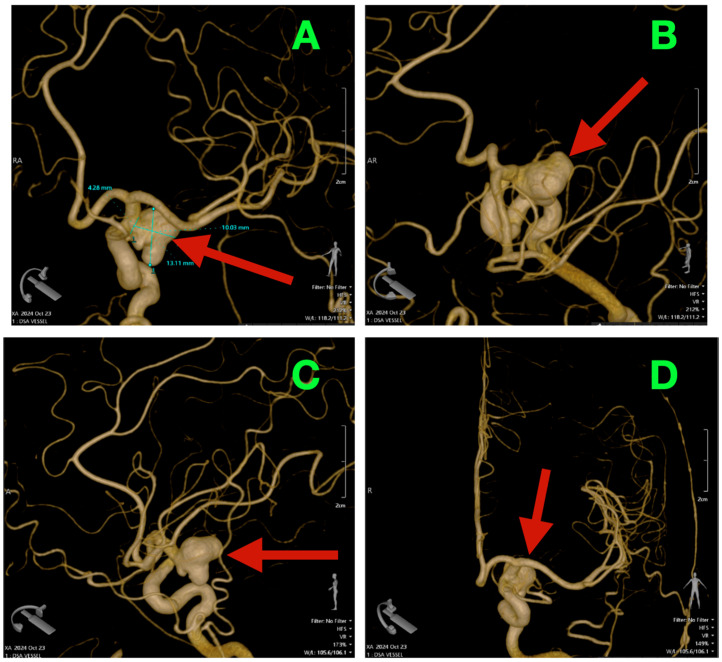
Three-dimensional rotational angiography with volume rendering and surgical measurements. (**A**): Quantitative reconstruction showing height 13.1 mm, width 10.0 mm, neck 4.3 mm (aspect ratio ≈ 3.06; dome-to-neck ≈ 2.34). (**B**): Oblique “working” view highlighting the posteriorly oriented, bilobed sac and its relationship to the carotid–optic window; the dome nests beneath the optic apparatus, forecasting a narrow margin for superior blade trajectory. (**C**): True lateral working view delineating the PCoA take-off and the posteromedial perforator bed abutting the neck, the critical zone to be visualized during neck dissection and clip deployment. (**D**): AP volume render situating the aneurysm within the opto-carotid cistern, illustrating the clearance required for clip passage while maintaining PCoA and anterior choroidal patency and avoiding posterior wall torque toward the perforators.

**Figure 3 diagnostics-16-00124-f003:**
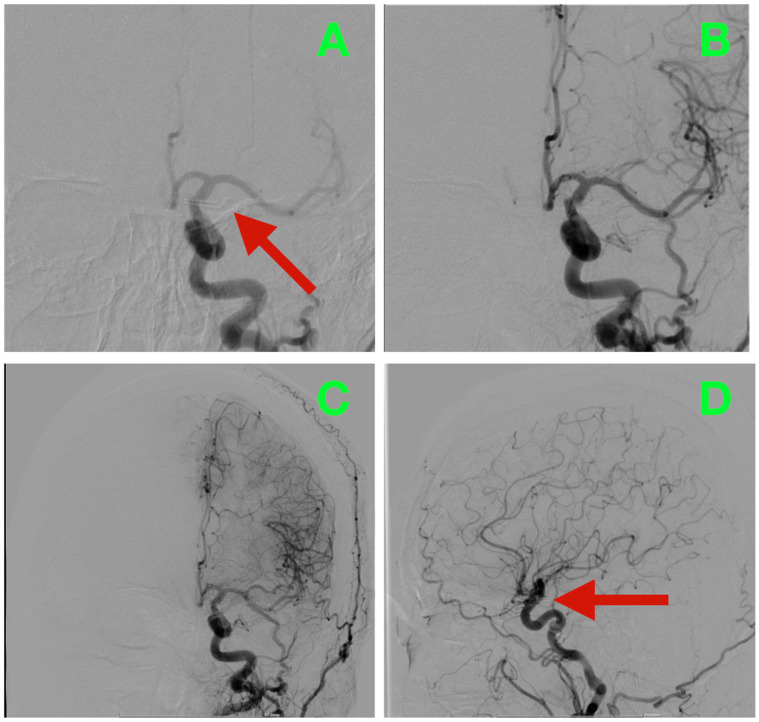
Early postoperative digital subtraction angiography of the left ICA. (**A**): AP projection: no residual filling at the ICA–PCoA junction; the clip line corresponds to the neck plane. (**B**): Oblique working view: caliber-preserved supraclinoid ICA with a free PCoA origin and smooth contour. (**C**): AP run: normal hemispheric opacification without pruning or delayed transit—no angiographic vasospasm. (**D**): Lateral projection: complete obliteration at the reconstruction site with an uncompromised anterior choroidal course.

**Figure 4 diagnostics-16-00124-f004:**
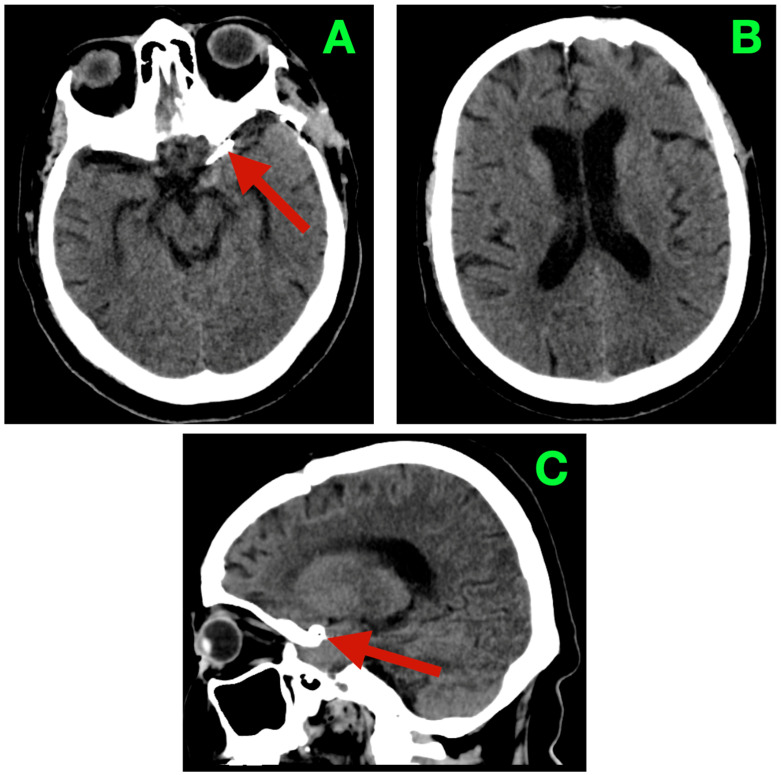
Three-month non-contrast cranial CT. (**A**): Axial basal slice: stable clip complex in the left suprasellar/carotid–optic cistern (arrow); basal cisterns patent; no hemorrhage. (**B**): Axial high-convexity slice: ventricular configuration normal with no ventriculomegaly or transependymal seepage. (**C**): Midline sagittal reconstruction: unchanged parasellar clip position (arrow), no extra-axial collection and no parenchymal hypodensity to suggest delayed ischemia.

**Table 1 diagnostics-16-00124-t001:** Selected studies from 2024–2025 that we found helpful for day-to-day decisions in posteriorly directed ICA–PCoA aneurysms. The table is not exhaustive; it highlights practical signals—what to measure on preoperative imaging, when to adjust the corridor, how outflow may affect device behavior, how to verify occlusion after clipping and how to gauge early DCI risk and system factors.

Domain and Question	Study and Year	Design and *n*	Key Anatomic or Physiologic Signal	Modality or Context	Principal Finding	Practical Translation for ICA–PCoA Cases Like Ours
Can we preflag difficult “low-lying” PCoA aneurysms on preop DSA rather than discovering them at arachnoid opening?	Ma et al., 2024, *Neurosurgical Review* [1]	Retrospective derivation with internal validation, *n* = 89 clipped PCoA aneurysms	Two simple 3D-DSA measures—a standard plane across both anterior clinoid tips and an ICA tortuosity angle—predict when the anterior petroclinoid fold and clinoid overhang the proximal neck	Microsurgical clipping only	A two-parameter DSA model accurately identified low-lying configurations and anticipated the need for APF fenestration or clinoid work to obtain proximal control and a posterior-neck sightline	Measure the overhang and tortuosity before incision; if low lying is predicted, plan early cisternal decompression and be ready for skull-base maneuvers to earn a direct posteromedial neck view.
When the pterional line of sight cannot show the posteromedial neck, does a subtemporal corridor rescue visualization and branch preservation?	Lan et al., 2024, *Frontiers in Neurology* [5]	Single-center original series of true or posteriorly projecting PCoA aneurysms	Direct lateral line provides early view of PCoA origin and perforators when retrocarotid sightline is blocked	Microsurgical clipping via subtemporal route	Reported complete clip occlusion with preservation of related branches in a small but focused cohort; authors emphasize selection for cases with constrained pterional sightlines	Treat approach as a line-of-sight problem; if the pterional window cannot expose the posteromedial neck safely, move laterally to protect perforators rather than force a blind closure.
Does fetal-type PCA alter PCoA aneurysm morphology and rupture risk in a way we can quantify preoperatively?	Han et al., 2025, *Journal of Clinical Medicine* [2]	Single-center radiomics cohort, *n* = 87 PCoA aneurysms	Presence of fPCA associates with shape irregularity and rupture status; combined radiomic–morphometric features outperform classic ratios	Mixed treated cohort; analysis at baseline imaging	Radiomics features plus fPCA status yielded better discrimination of rupture than size-based indices alone	When fPCA is present, assume altered neck inflow/outflow geometry and a higher likelihood of irregular domes; insist on direct posteromedial visualization before committing a clip line.
In flow diversion at the PCoA, how does fetal posterior circulation change device performance?	MacRaild et al., 2025, *Journal of NeuroInterventional Surgery* [3]	Original in silico assessment across many individualized PCoA anatomies	Fetal circulation reduces postdevice velocity suppression and increases device surface shear	Off-label PED scenarios modeled computationally	Predicted less effective hemodynamic quenching and slower endothelialization surrogates in fPCA configurations compared with adult outflow	If robust fPCA outflow persists, be cautious with single-device strategies; consider exact neck reconstruction when branch incorporation and back-wall access permit.
After clipping, can 3D-CTA replace DSA to detect clinically important remnants?	Image control after aneurysm-clipping study, 2025, *Neurocirugía* [14]	Prospective comparative evaluation	Modern 3D-CTA detects many small remnants but has sensitivity limits for tiny or clip-shadowed necks	Early postoperative imaging	DSA remained decisive for indeterminate findings and for small residual necks that inform reintervention decisions	Use 3D-CTA to reduce invasiveness when the construct is unequivocal; reserve DSA as reference when clip angles or branch run-off are even mildly uncertain.
Do temporary clips leave parent-vessel changes that are microscopically occult?	Hendrix et al., 2024, *Journal of NeuroInterventional Surgery* [13]	Hybrid-OR series with intraoperative diagnostic cerebral angiography (ioDCA)	ioDCA reveals vasospasm or luminal irregularity after temporary occlusion not apparent under the microscope	Elective aneurysm clippings with ioDCA	Demonstrated angiographic sequelae attributable to temporary clipping despite unremarkable microscopic appearance	Keep proximal quieting brief and purposeful; if occlusion was anything beyond short and bloodless, verify with angiography before closing.
If a pinpoint remnant is seen intraoperatively, can ioDSA flow behavior predict spontaneous thrombosis?	Grüter et al., 2024, *Acta Neurochirurgica* [12]	Original cohort of clipped aneurysms with ioDSA-detected remnants	Intra-aneurysmal contrast stasis during ioDSA predicts later remnant thrombosis	Microsurgical clipping with ioDSA	Remnants showing contrast stasis were more likely to thrombose without reintervention on follow-up imaging	When ioDSA shows stasis in a minute remnant and branch run-off is pristine, observation can be reasonable instead of risky clip revision.
Can we predict DCI at admission using only non-contrast CT features?	Chen et al., 2025, *BMC Medical Imaging* [15]	Retrospective multicenter radiomics nomogram, *n* = 377 aSAH	Quantitative NCCT features stratify early DCI risk without contrast or perfusion	Standardized baseline CT	NCCT radiomics nomogram achieved robust early DCI prediction and outperformed clinical scales alone	Use imaging priors to tailor surveillance intensity and imaging cadence while maintaining euvolemia and nimodipine exposure.
Do combined clinical, radiomics and deep-learning features improve DCI prediction beyond bedside scores?	*AJNR* 2024 multicenter model [16]	Original predictive study integrating three data layers	Fusion of clinical data with radiomics and deep neural features improves prediction stability	Development and internal validation	Best-performing models outpaced clinical-only baselines for DCI discrimination and calibration	Consider structured collection of baseline imaging for centers building DCI risk stratification pipelines.
What are the system-level economics and trends in aSAH care in a middle-income country?	*Frontiers Neurology* 2025, Brazil nationwide study 2017–2022 [17]	National database time-series, >61,000 aSAH admissions	Costs and procedure patterns during and after the pandemic	Real-world health-system analysis	In-hospital mortality ~20% and rising absolute costs with ICU days and procedures as primary drivers	Planning for modality and timing must account for ICU capacity and cost; standardized pathways reduce delays and waste.
Does center volume still track outcomes for ruptured aneurysms in a mature EVT era?	*JNIS* 2025, Germany national analysis [6]	Nationwide real-world cohort across hospitals	Annual center volume as an outcome modifier	Clipping and EVT for ruptured IAs	Higher-volume centers achieved better survival and independence after adjustment for case-mix	Regionalize complex microsurgery and ensure rapid transfer when ICA–PCoA geometry mandates clip-based reconstruction.
Which variables prolong length of stay and drive disposition after aSAH?	*Neurocirugía* 2024, LOS and LTCF predictors [18]	Original cohort analysis	Neurological grade, complications and social factors dominate LOS and discharge destination	Mixed modality aSAH care	Modality per se contributed less to LOS than complications and initial grade	Highest cost is in days in care; preventing hydrocephalus, DCI and infections often matters more economically than the index technique.

## Data Availability

The data presented in this study are available on request from the corresponding author. The data are not publicly available due to privacy and ethical restrictions related to patient confidentiality.

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
