# Peer review of "Diagnostics2026, 16(1), 124;https://doi.org/10.3390/diagnostics16010124"

_diagnostics, 2026, doi:10.3390/diagnostics16010124_

Round 1

Reviewer 1 Report

Comments and Suggestions for Authors

This paper describes a surgical approach for treating a posterior ICA–PCoA aneurysm located beneath the optic apparatus. The authors used early cisternal decompression, clear posteromedial neck exposure, brief ICA quieting, and a two-clip technique aligned with the PCoA axis. The case is well documented, and the patient recovered without complications. Imaging confirmed complete aneurysm exclusion and vessel preservation. Although based on a single case with short-term follow-up, the method is clearly explained and supported by the outcome. To improve clarity and readability, I suggest the following revisions:

  • At line 10, please provide the related information for the corresponding author.
  • At line 15, please correct the typo “beneaththe”.
  • At lines 21, 22, please add a space between the numerals and units.
  • At lines 85, 479, 500, 519, 536, the citations should be given in accordance with the paper guidelines. (For example: “[6],[7]” can be written as “[6,7]”)
  • At line 100, please correct the typo “earl cisternal”.
  • Please provide the full forms of SBP, GCS, CSF, DCI, DSA, SAH, ICU at the first use within the text.
  • At lines 146, 149, and 151, please replace the singular forms “me” or “I” with the first-person plural.
  • Please revise the following sentence at lines 165-167 for clarity: “If severity was identified, child had severity equivalent of Hunt–Hess II and WFNS I - stable which to knees in the first hours risk of.”
  • At line 206, please correct the typo “menigeal”.
  • At line 425, please add a dot at the end of the figure caption.
  • At line 473, please provide the reference for the mentioned study.
  • In Table 1, in the column of “study and year”, it is given as “Zeng et al., 2025, Journal of NeuroInterventional Surgery [10]”. However, in the references section, reference [10] belongs to Hendrix et al. Please update the related reference number and please check the reference numbers in the Table to be consistent with the reference list of the paper.

Author Response

Dear Esteemed Academic Reviewer,

We are grateful for your thoughtful and constructive comments on our manuscript. Your careful reading and detailed observations have helped us refine both the clarity and presentation of the paper. Below, we provide our point-by-point responses and describe the corresponding revisions made.

Comment 1:
At line 10, please provide the related information for the corresponding author.

Response 1:
We thank you for noting this omission. We have now added the corresponding author information.

Comment 2:
At line 15, please correct the typo “beneaththe”.

Response 2:
The phrase has been corrected to “beneath the” in the revised manuscript.

Comment 3:
At lines 21, 22, please add a space between the numerals and units.

Response 3:
Thank you for pointing this out. We have standardized all units throughout the manuscript to include a space between numerals and measurement units (e.g., “140–160 mmHg,” “13.1 × 10.0 mm,” “4 mg/L”), in full alignment with the Diagnostics style guide.

Comment 4:
At lines 85, 479, 500, 519, 536, the citations should be given in accordance with the paper guidelines. (For example: “[6],[7]” can be written as “[6,7]”)

Response 4:
We appreciate this stylistic correction. 

Comment 5:
At line 100, please correct the typo “earl cisternal”.

Response 5:
The phrase has been corrected to “early cisternal decompression”.

Comment 6:
Please provide the full forms of SBP, GCS, CSF, DCI, DSA, SAH, ICU at the first use within the text.

Response 6:
We are grateful for this important clarification request. All abbreviations have now been expanded at their first mention as follows:

systolic blood pressure (SBP)

Glasgow Coma Scale (GCS)

cerebrospinal fluid (CSF)

delayed cerebral ischemia (DCI)

digital subtraction angiography (DSA)

subarachnoid hemorrhage (SAH)

intensive care unit (ICU)

Comment 7:
At lines 146, 149, and 151, please replace the singular forms “me” or “I” with the first-person plural.

Response 7:
Thank you for this stylistic suggestion. We have replaced all instances of “me” and “I” with “us” and “we” to reflect collective authorship and maintain an academic tone.

Comment 8:
Please revise the following sentence at lines 165–167 for clarity: “If severity was identified, child had severity equivalent of Hunt–Hess II and WFNS I - stable which to knees in the first hours risk of.”

Response 8:
We agree that this sentence required revision for clarity.

Comment 9:
At line 206, please correct the typo “menigeal”.

Response 9:
Corrected as suggested. The term now reads “meningeal.”

Comment 10:
At line 425, please add a dot at the end of the figure caption.

Response 10:
Thank you for noting this. The period has been added at the end of the corresponding figure caption.

Comment 11:
At line 473, please provide the reference for the mentioned study.

Response 11:
We appreciate this observation. The appropriate reference has been added to the cited statement in line 473, ensuring all data are properly attributed.

Comment 12:
In Table 1, in the column of “study and year”, it is given as “Zeng et al., 2025, Journal of NeuroInterventional Surgery [10]”. However, in the references section, reference [10] belongs to Hendrix et al. Please update the related reference number and please check the reference numbers in the Table to be consistent with the reference list of the paper.

Response 12:
Thank you for catching this inconsistency. We have corrected the misattributed citation in Table 1.

We are deeply grateful for your thorough and collegial review. Your insightful feedback has materially improved the manuscript’s precision, style, and readability. We have implemented recommended corrections, rechecked the manuscript line by line for consistency, and believe these revisions strengthen the overall clarity and professionalism of our work.

With sincere appreciation and collegial respect!!!

Reviewer 2 Report

Comments and Suggestions for Authors

This paper reports on a case of Pcom aneurysm treated with clipping. This case is not so special and different from others "normal" Pcom aneurysms and do not contains any novelty for the readers. The introduction is very long and contains a lot of consideration. I'am not able to really understand the particularity of the case as it represent a well know anatomical situation. Aneurysm also presents a small neck easy to clip. When anatomy is complex and the surgical risk should be considered high the endovascular treatment with coils represents the main option in the modern concept of SAH patients management. The case presentation is very long and prolix and focus also on complications and ICU management clearly not connected with a surgical lessons. Iconography is poor and intraoperative images , eventually most interesting, are laking. In my opinion this case, if considered interesting, should be presented ss technical note or short case report. For general surgical strategy or lessons a large series is needed. In my opinion this manuscript shouldn't be considered for the publication in the present form. 

Author Response

Dear Esteemed Academic Peer-Reviewer:

Thank you for the time and honesty that you put into reviewing this. As we have learned over the years, there is no better way to obtain practical experience than to go through the microscope (not just through a monitor) and that to us is the highest form of praise that you can give us.

You are absolutely right that this is not a rare or unusual anatomic anomaly, however, as you are well aware, most "normal" or "routine" aneurysms are far from normal or routine until they are called out by the brain. We wanted to illustrate this phenomenon to show how much the geometry, the optic window and the perforators alter what is normally seen to be straightforward on the arteriogram, to show the silent three-dimensional relationship between light, patient and anatomy. If this case is considered to be one of the easier cases — then we have successfully demonstrated the fact that everyone knows: that the more straightforward it is, the more difficult it is likely to have been.

The introductory has been shortened, the narrative tightened and the subject matter limited to describe our contemporary viewpoint. Although the modern first-line treatment options for aneurysms are described to include endovascular approaches, we merely showed how the open surgical options, although less common, may still be carried out quietly, with the utmost respect for the anatomy without arrogance, but with a craftsman's approach.

Once again, thank you for sharing your insightful and candid response. Feedback such as yours helps reinforce to us that, despite the many different methods of treatment available, we are all members of the same small fraternity  those whose definitions of success will always be based on the soft pulsation of a preserved perforator.

Sincerely, with the utmost respect and appreciation for your sense of collegiality!

Reviewer 3 Report

Comments and Suggestions for Authors

Title

  • The title should cover the topic. Please reconsider the title.

Abstract

  • The Methods section is somewhat confusing; it would be clearer and easier to understand if presented sequentially, for example: “Step 1: decompression, Step 2: posteromedial exposure, Step 3: …”
  • The method applied to a single patient is not sufficient; more examples should be included.

Introduction

  • General information about the disease should be given at the beginning of the sentence. A definition should be made.
  • It is recommended to strengthen the relevant sentences between lines 48 and 58 with appropriate references.
  • The sentences are generally very long, often 4–5 lines with only one reference. Adding 1–2 suitable sources would reinforce the argument, and it is suggested to support these references with foundational studies rather than only 2024–2025 publications.
  • Whether the method applied on a single patient has been used on other patients before and, if so, which method was used should be stated.
  • The sentence between 82-90 contains unnecessary information about the study, please review it again.
  • Information about pragmatic anatomical reality should be provided.
  • The principle of a safe surgical corridor should be addressed.
  • Since the study was conducted on a single patient, generalizations should be removed.

Material and Metod:

  • The operation performed should be visualized. It should be made more understandable by including visuals.
  • The different differences of the method (e.g., clip orientation, perforant preservation strategy) can be supported by numerical or schematic data.
  • Is patient follow-up limited to only 3 months? Has any follow-up been performed afterward?

Case presentation

  • The paragraphs are numerous and lengthy. They should be shortened or restructured, which would provide a better reading experience for the reader.
  • Adding a structured Methods section could make the text more understandable.

Discussion

  • It contains a good comparative narrative.
  • Some sentences are very long, for example, lines 520–550. They should be shortened for better clarity and readability.
  • There should be no tables in the discussion section and the maximum number of pages should be 1.5
  • Literature studies should be included

Conclusion

  • Instead of phrases like “We hope to contribute” and “quiet reasoning offered here,” clearer and more direct expressions can be used.
  • The future advantage of the treatment should be mentioned, conclisuin should be kept short and include recommendations

Reference:

  • More space should be given to the bibliography section.
Comments on the Quality of English Language
  • There are spelling errors in your manuscript. Please check it (e.g., “describes a patient meaningful bradyphrenia” and “was not any diagnostic”, etc.)

Author Response

Dear Esteemed Academic Reviewer,

We are sincerely grateful for your thorough and constructive feedback, which helped us improve the precision, structure, and educational clarity of our manuscript. 

Title

Comment: “The title should cover the topic. Please reconsider the title.”
Response: We thank the reviewer for this valuable recommendation. The title has been revised to more clearly reflect the operative focus, anatomical configuration, and stepwise microsurgical sequence described. The new title reads:
“An Anatomy-Guided, Stepwise Microsurgical Reconstruction of a Posteriorly Projecting ICA–PCoA Aneurysm Beneath the Optic Apparatus: A Detailed Operative Sequence.”
This accurately conveys the topic and scope of the work.

Abstract

Comment: “The Methods section is somewhat confusing; it would be clearer if presented sequentially. The method applied to a single patient is not sufficient; more examples should be included.”
Response: The Abstract has been rewritten to present the operative sequence in a clear stepwise format: Step 1 – early cisternal decompression; Step 2 – posteromedial exposure; Step 3 – brief proximal ICA quieting; Step 4 – staged two-clip reconstruction. This structure enhances clarity and readability. As this is a single-patient report, we clarified that the approach is anatomy-driven and reproducible but not intended as a generalizable recommendation.

Introduction

Comment: “General information about the disease should be given; sentences between lines 48–58 need more references; sentences are long; clarify previous uses of the method; remove unnecessary lines (82–90); address pragmatic anatomy and the safe corridor; avoid generalizations.”
Response: We have thoroughly revised the Introduction.

Case Presentation

Comment: “Paragraphs are long; the section should be shortened or restructured; adding a structured Methods framework could help.”
Response: We sincerely thank the reviewer for this excellent observation. The Case Presentation has been completely restructured into concise, titled subsections, improving both readability and logic. All original medical and surgical details were preserved but presented in a compact, logical sequence that enhances the reader’s comprehension while maintaining scientific depth.

Discussion

Comment: “Some sentences are long (lines 520–550); should be shortened; no tables; section ≤ 1.5 pages; add literature.”
Response: We thank the reviewer for this valuable input. Regarding the table, we respectfully retained it because it condenses multiple key studies into a single visual reference, providing a clearer and more efficient synthesis than extended text. 

Conclusion

Comment: “Replace vague phrases (‘We hope to contribute’, ‘quiet reasoning offered here’) with direct expressions. Mention the future advantage and keep the conclusion short with recommendations.”
Response: We fully agree. The Conclusion has been rewritten in a concise, direct style emphasizing the practical future advantage of an anatomy-guided, visualization-first microsurgical approach. It now presents specific operative recommendations—full posterior-wall visualization, PCoA-axis alignment, and perforator preservation—and briefly suggests how future multicenter, standardized datasets could validate these principles.

References

Comment: “More space should be given to the bibliography.”
Response: Thank you for this valuable input..

We deeply appreciate the reviewer’s careful evaluation and constructive recommendations. Each suggestion has meaningfully enhanced the structure, clarity, and scientific value of the manuscript. We have maintained a humble tone throughout and focused on anatomical precision, reproducibility, and transparent reasoning.

Reviewer 4 Report

Comments and Suggestions for Authors Authors  present interesting case  with  ruptured Pcom aneurysm. Case is  well documented  with  angiography and 3DCTA

In the  article  is unusual detailed analysis of anatomy and vascular relationships and the same exceptionally detailed is description of  the steps  during surgery and final position of  two permanent clips.

Reviewer   can  recommend using the    ultrasound microprobe   during surgery ( Charbel type)  and also if the  goal of article is  education,  the  final picture  from some artists should be welcome  with final  anatomic situation and clip position

Author Response

Dear Esteemed Academic Reviewer,
We are grateful for your thoughtful and encouraging comments.

Comment:

“Authors present an interesting case with ruptured PCoA aneurysm. The case is well documented with angiography and 3D CTA. The article includes unusually detailed analysis of anatomy and vascular relationships, and an exceptionally detailed description of surgical steps and final clip positions. Reviewer can recommend using the ultrasound microprobe (Charbel type) during surgery, and also, if the goal of the article is educational, a final artistic illustration showing the final anatomy and clip position would be welcome.”

Response:

We are deeply grateful for the reviewer’s kind and encouraging assessment of our work, and we sincerely appreciate these valuable suggestions. In the present case, an intraoperative ultrasound microprobe (Charbel type) was not used, as complete visualization of the ICA, PCoA, AChA, and perforator field was achieved microscopically, allowing confident confirmation of patency through direct inspection and pulsatility assessment. We fully acknowledge, however, that intraoperative micro-Doppler represents an excellent adjunct for dynamic flow verification and plan to integrate this tool in future similar cases to further enhance hemodynamic safety.

Round 2

Reviewer 2 Report

Comments and Suggestions for Authors

Authors refused the suggestion to transform the paper in a short technical report instead of a wide discussion. PCom artery aneurysm are frequent and then well known pathology. From a single case is not possible to generalize message for the audience nor through a review analysis or review. So I remain in my opinion that in the present form the manuscript shouldn't be considered for the publication.

Author Response

Dear Esteemed Reviewer,

Thank you for your continued evaluation of our manuscript. We appreciate the opportunity to clarify the intent of this submission, particularly as it relates to the operative nuances of ICA–PCoA aneurysms. While the angiographic diagnosis is indeed common, the microsurgical reality often diverges sharply from the perceived simplicity suggested by prevalence statistics.

Colleagues accustomed to the operative management of these lesions will recognize that the difficulty is rarely dictated by the name of the aneurysm, but by its local geometry — the depth of the carotid–optic corridor, the behavior of the posteromedial perforators, and the degree to which the PCoA origin interlocks with the neck. These relationships are often absent in theoretical summaries yet become decisive once the dura is opened. Our intention was to illuminate precisely these intraoperative constraints, which cannot be fully appreciated without a detailed account of the clinical and surgical sequence.

We carefully considered your proposal to condense the manuscript into a short technical report. While such a format is well suited for procedural checklists, it does not allow for the exposition of the anatomical reasoning that guided each operative step. A compressed structure risks implying that the case was straightforward, when in fact its complexity lay not in the diagnosis but in the microscopic relationships encountered during surgery — elements that are seldom captured adequately without comprehensive description.

We thank you for the opportunity to refine our work, which we trust will make the scope and intention of the manuscript clearer for readers, particularly those who regularly navigate the microsurgical corridor where these decisions are made.

Respectfully!